

# Variation pattern of particulate organic carbon and nitrogen in oceans and inland waters

Changchun Huang[1,2,3,*], Lin, Yao[4], Hao Yang[1,3], Chen Lin[5], Tao Huang[1,3], Mingli Zhang[1], A-xing Zhu[1,2,6],Yimin Zhang[3]

5    1 Jiangsu Center for Collaborative Innovation in Geographical Information Resource Development and Application, Nanjing Normal University, Nanjing 210023, China

2 Key Laboratory of Virtual Geographic Environment (Nanjing Normal University), Ministry of Education, Nanjing 210023, China

3 School of geography science, Nanjing Normal University, Nanjing 210023, China

4 State Key Laboratory of Resources and Environmental Information System, Institute of Geographic Sciences and Natural Resources Research, Chinese
10   Academy of Sciences

5 Key Laboratory of Watershed Geographic Sciences, Institute of Geography and Limnology, Chinese Academy of Sciences, Nanjing 210008, China

6 Department of Geography, University of Wisconsin, Madison, Wisconsin 53706, USA

*Correspondence to:* Changchun Huang (huangchangchun_aaa@163.com)

**Abstract:** We examined the relationship between, and variations in, particulate organic carbon (POC) and

15   particulate organic nitrogen (PON) based on previously acquired ocean and inland water data. Some new points

were found beside the traditional latitude, depth and temperature dependence of POC, PON and POC/PON. The

global average value of POC/PON (7.54±3.82) is higher than the Redfield ratio (6.63). The mean values of

POC/PON in south and north hemisphere are 7.40±3.83 and 7.80±3.92, respectively. The high values of

POC/PON appeared between 80° N ~ 90° N (12.2±7.5) and 70° N ~ 80° N (9.4±6.4), and relatively low

20   POC/PON were found from 20 °N (6.6±2.8) to 40 °N (6.7±2.7). The latitudinal dependency of POC/PON in the

northern hemisphere is much stronger than in the southern hemisphere. Variations of POC/PON in inland water

also showed similar latitude-dependency of POC/PON in ocean water, but significantly regulated by lake's

morphology, trophic state and climate, etc. factors. Higher POC and PON could be expected in the coastal water,

while POC/PON significantly increased from 6.89 ± 2.38 to 7.59 ± 4.22 in north hemisphere with the increasing

rate of 0.0024/km. The coupling relationship between POC and PON in oceans is much stronger than in inland

waters. Variations in POC, PON and POC/PON in inland waters should receive more attention due to the

importance of these values to global carbon and nitrogen cycles and the indeterminacy of the relationship

between POC and PON.

# 1. Introduction

Inland waters and oceans transport, transform and contain large amounts of organic carbon and thus play an important

role in global carbon, nitrogen and nutrient cycles (Cole et al., 2007). Inland waters receive carbon from terrestrial

ecosystems at a rate of 2.9 PgC/yr. Of this carbon, 21% (0.6 PgC/yr) is stored in sediment, 48% (1.4 PgC/yr) is emitted

to the atmosphere as $CO_2$, and 31% (0.9 PgC/yr) is discharged into oceans via rivers (Tranvik et al., 2009). The ocean

is an important carbon sink due to the flux of riverine carbon (0.9 PgC/yr) and the absorption of atmospheric $CO_2$,

which is fixed by phytoplankton at a rate of 2.3±0.7 PgC/yr (IPCC, 2013).

There is a strong relationship between nitrogen and carbon cycles in natural aquatic ecosystems. The input of nitrogen

into aquatic ecosystems as nutrients from the land and atmosphere stimulates additional uptake of carbon (Hyvönen et

al., 2007), and fixed carbon and nitrogen are released as gas and ions ($CO_2$, $CH_4$ and NOx, etc.) when organisms are

mineralized (Galloway, et al., 2004; Flückiger et al., 2004). This relationship is made stronger by the life processes of

organisms, but it is weakened by variations in the sequestration and mineralization rates of carbon and nitrogen

(Gruber and Galloway, 2008). The relationship between carbon and nitrogen is relatively stable in natural aquatic

ecosystems, although carbon and nitrogen levels vary depending on autotrophic biotypes and water environment

(Thornton and McManus, 1994; Gruber and Galloway, 2008). This relationship is also affected by human activity

(Gruber and Galloway, 2008; Galloway et al., 2008; Perga et al., 2016).

The elemental composition of organic matter affects the global biogeochemical cycle and varies depending on its

sources (DeVries and Detsch, 2014). The carbon to nitrogen (C/N) ratio affected by the life processes of organisms and

is a good measure of the relationship between carbon and nitrogen cycles (Sterner and Elser, 2002; Schneider et al.,

2003; Meisel and Struck, 2011; Babbin et al., 2014). Organic nitrogen originates from plant proteins and nucleic acids

and, to a lesser extent, from lignin and cellulose. The C/N ratio in terrestrial plants is much higher than in autotrophic

phytoplankton due to their high lignin and cellulose content (Kendall et al., 2001; Mcgroddy et al., 2004; Watanabe

and Kuwae, 2015). This leads to a C/N ratio that is higher and much more variable in inland waters than in offshore

oceans; there is also a sharp contrast in nutrient levels and water residence times between the two (Hall et al., 2007,

Sterner et al., 2008; Watanabe and Kuwae, 2015). Several studies suggest that the currently observed C/N ratio, and

variations in it, are difficult to reconcile with the value estimated by Redfield (6.63-7.7), which was based on data

taken from ocean-surface plankton and deep, dissolved nutrients from 1898 to 1933 (Kokrtzinger et al., 2001;

Schneider et al., 2004; Koeve, 2006; Sterner et al., 2008; Martiny et al., 2013a; 2013b; DeVries and Deutsch, 2014;

Watanabe and Kuwae, 2015). The factors influencing variations in C/N are complex; nitrogen and light limitation and

phytoplankton can only explain approximately 20% of the variation in C/N on a global scale (Martiny et al., 2013b).

Other factors that regulate C/N on a regional scale include microzooplankton (Talmy et al., 2016), heterotrophic

microbes (Crawford et al., 2015) and terrestrial organisms (Jiang, 2013). This variation in C/N increases the

uncertainty of global carbon and nitrogen estimation (Babbin, 2014). Consequently, understanding temporal and spatial

variation in particulate organic carbon (POC), particulate organic nitrogen (PON) and the POC/PON ratio, as well as

the processes that govern POC/PON, is critical to better explain the global biogeochemical cycles of carbon and

nitrogen.

Recently, global oceanic studies have proposed that the median global value of C/N in oceans is close to the Redfield

value, but there is significant regional variation (Martiny et al., 2013b). Meanwhile, POC/PON exhibits a strong

latitudinal pattern, with lower values in the cold ocean waters of the higher latitudes (Martiny et al., 2013a). In contrast

to the study of oceanic POC/PON, the elemental stoichiometry research of C/N in inland waters is still need to be

complement and perfection (Sterner et al., 2008). In this study, we extend the study area and dataset of previous studies

(Martiny et al., 2013b; 2014; Kim et al., 2015), from 60° N ~ 78° S with 40482 samples to 80° N ~ 78° S with 63184

samples, to re-examine variations in POC, PON and POC/PON on a global scale. Values for POC, PON and POC/PON

in inland waters were combined to further reveal the relationship between POC and PON and deviations in C/N from

the classical Redfield value.

**2. Data and Methods**

2.1 Data collection

To achieve this study's objective, datasets from previously published studies and publicly available online data were acquired (detailed information was listed in the supporting material Table S1). This compiled dataset contained 63,184 paired POC and PON samples (northern hemisphere, 40,809 samples; southern hemisphere, 22,448 samples) from offshore and coastal oceans and 23,996 samples from inland waters (rivers and lakes). The spatial distribution of samples is shown in Figure 1. Measurements of particulate elements were carried out by standard methods, which C and N were analyzed on C/N elemental analyzer after water samples filtered through preweighed, precombusted (450°C for 4 hours) GF/F filters and acidified treatment. Some corrections for inorganic carbon and nitrogen were implemented, but we lack the detailed information on correction effect and how large influence of inorganic matters although the filtered samples were treated by acidified. The units of POC and PON in all data (μg/L, μm/L) were unified to um/L via molecular-weight of C and N.

Geographical land and ocean distribution data and soil organic carbon data (harmonized world soil database, http://daac.ornl.gov/SOILS/guides/HWSD.html) were also used to analyze the factors influencing variations in POC, PON and POC/PON.



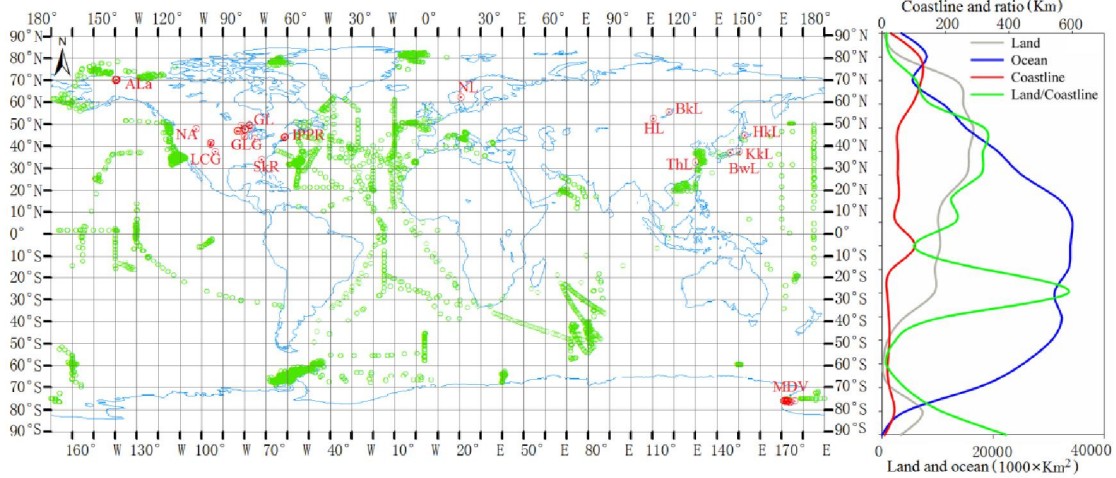

Figure 1 Global distribution of paired samples of POC and PON. Green circles are oceanic samples and red circles are inland samples. Land and ocean area, coastline length and the ratio of land area to coastline are also shown. The original map data of world vector downloaded from http://www.naturalearthdata.com/.

## 2.2 Data category and analyses methods

Samples with extremely low POC (< 0.01 μm/L) and PON (< 0.001 μm/L) values were removed due to the limit

of detection for the analysis. The collection data of ocean from previous studies and web-sharing database

contained variable ranges of POC and PON with latitude, time, depth and temperature. In order to reveal the

pattern of POC and PON, the remaining data were classified into groups relating to latitude, depth, offshore

distance and temperature to aid data analysis: 1) Oceanic POC and PON data, taken from 80° N to 78° S, were

divided into 17 ranges with 10° latitudinal intervals (Table S2). The data in each latitudinal range include all

ranges of temperature, time and depth for POC and PON. 2) Ocean POC and PON samples were separated into

0-5m, 5-10m, 10-20m, 20-80m and >80m depth ranges according to the distribution of samples' number (Table

S3 and Figure S1). The distribution of samples in each depth covers most ranges of latitude, time and temperature

for POC and PON. 3) Offshore distance ranges (5, 10, 25, 50, 100, 200 and 300 km) were created via Arcgis 10

(Esri), and the samples located in each range were separated and statistically analyzed (Table S4). All ranges of

POC and PON with different depth, time and latitude in each buffer were treated as same category. 4) POC and

PON samples were divided into temperature ranges with an interval of approximately 1 degree centigrade.

The lake collection data of POC and PON was analyzed for individual lakes. Some measurements of POC and

PON are multiple observations in many small lakes. These observational data were processed as lake groups,

such as Great Lakes Group, Lacustrine Central Group and Alaskan Lakes (Table S6).

The number of samples was listed in the tables of supporting material (Table S2 - S5). Statistical values (mean,

maximum, minimum and standard deviation) were calculated for POC, PON and POC/PON for all groups. The

relationships between PON and POC for all categories (ocean and lakes) were also regressed and listed in the

supporting material. The relationships between particulate organic matter (PON and POC) and water properties

(temperature, DOM, chlorophyll and total suspended sediment), as well as the soil organic carbon, were regressed

to explore the effect of each influencing factors to the variation pattern of POC and PON.

## 3. Results and discussion

3.1 Latitude- and depth-dependent POC, PON and POC/PON variation in the ocean

The spatial distributions of POC, PON and POC/PON significantly affect marine carbon and nitrogen flux

estimation as well as the air-ocean exchange of $CO_2$ via the global ocean carbon cycle model (Schneider et al.,

2004). Studies have proposed that the elemental ratio (POC/PON) of particulate organic matter in marine

environments is characterized by a strong latitudinal pattern (for 60° N ~ 60° S) due to the influence of nutrients,

temperature and respiration (Martiny et al., 2013a; Devries and Deutsch, 2014). Microzooplankton and algae

production also regulate POC/PON in the ocean (Tamelander et al., 2013; Crawford et al., 2015; Talmy et al.,

2016). PON and POC co-vary, resulting in a strongly coupled relationship. Both POC and PON show a latitudinal

pattern globally, but variations in POC and PON in the northern hemisphere are much more variable than in the

southern hemisphere (Figure 2A, B); the latitudinal dependency of POC/PON in the northern hemisphere is much

stronger than in the southern hemisphere (Figure 2C). In contrast to a previous study (Crawford et al., 2015),

which observed that a low POC/PON ratio (2.1 to 5.6) existed in the middle-high latitudes (80° N ~ 50° N) due to

the presence of heterotrophic microbes in summer time, we found high values for POC/PON, $12.2\pm7.5$ and $9.4$

$\pm6.4$ between 80° N ~ 90° N and 70° N ~ 80° N, respectively. Relatively low POC/PON ratios were found from

20 °N ($6.6\pm2.8$) to 40 °N ($6.7\pm2.7$). Consistent with earlier studies, the low POC/PON ratios were very close to

the Redfield value (6.625). The determined coefficient ($R^2$) of the relationship between POC and PON in the

southern hemisphere is slightly higher than in the northern hemisphere (Table S2). The mean value of POC/PON

in northern hemisphere ($7.50\pm4.65$) is slight lower than in southern hemisphere ($7.81\pm3.79$). These indicate that

geobiochemical processes and the circulation of carbon and nitrogen in the northern hemisphere are much more

complex than in the southern hemisphere. The variation for POC/PON in the northern hemisphere is bigger than

in the southern hemisphere (Table S2).

Linear functions (including and excluding intercepts) and power functions can be used to express the relationship

between carbon and nitrogen for each latitudinal range (Table S2). However, the optimal function is different for each latitudinal range. The best regression results with the highest $R^2$ are noted with an asterisk in Table S2. A power function is used to describe the relationship between carbon and nitrogen globally (POC = (6.998±0.645) × PON$^{(0.901±0.081)}$, $R^2$ = 0.905 ± 0.052). A linear function (including and excluding intercepts) also describes the relationship between carbon and nitrogen well (POC = (7.545±2.498) × PON + (0.302±2.658), $R^2$ = 0.891 ± 0.047; POC = (7.666±2.169) × PON, $R^2$ = 0.882 ± 0.048). The regression functions for POC and PON are listed in Table S2. The slopes of the linear regressions in this study are bigger than in previous regional studies (e.g., 5.89, 5.06 and 4.63, Caperon, 1976; 6.43, Verity, 2002; 5.8, Lara et al., 2010; 5.53 and 5.38, Crawford et al., 2015; 6.62, Cai, P.H. et al., 2015; 6.75, Kim et al., 2015). However, the global mean value of POC/PON (7.54 ± 3.82) is higher than the Redfield ratio of 6.63, as well as some recent studies (6.62, Cai, P.H. et al., 2015; 6.75, Kim et al., 2015).



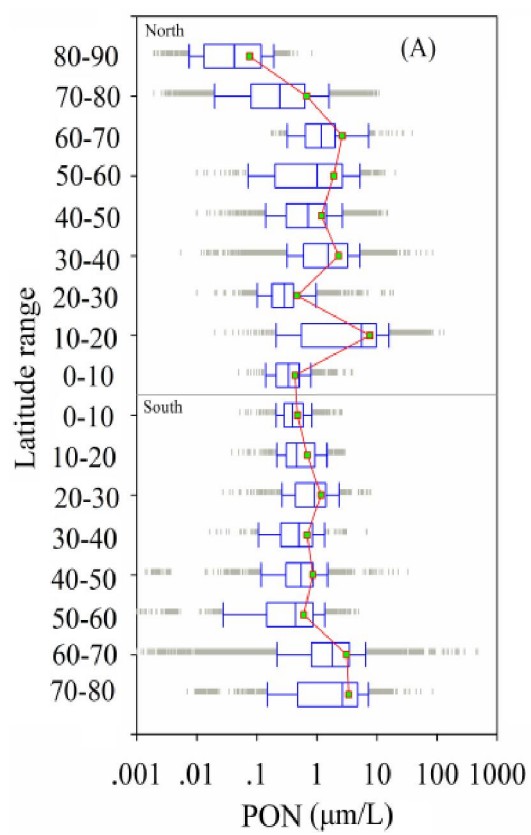

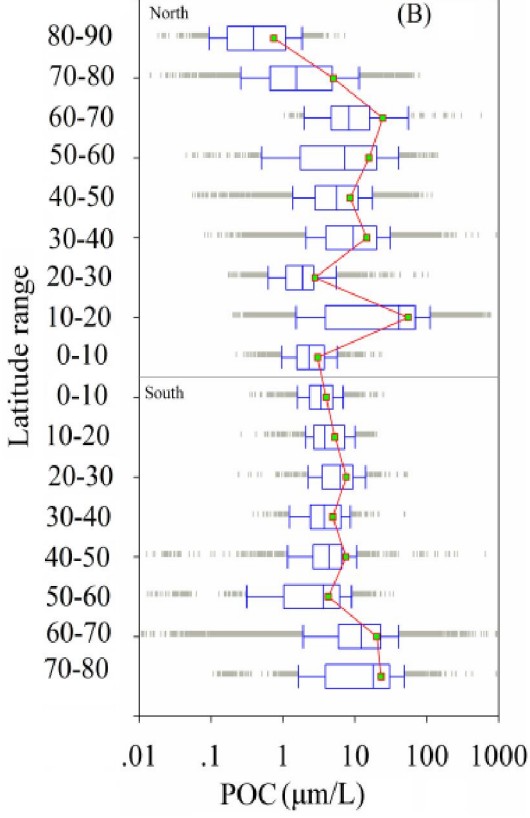





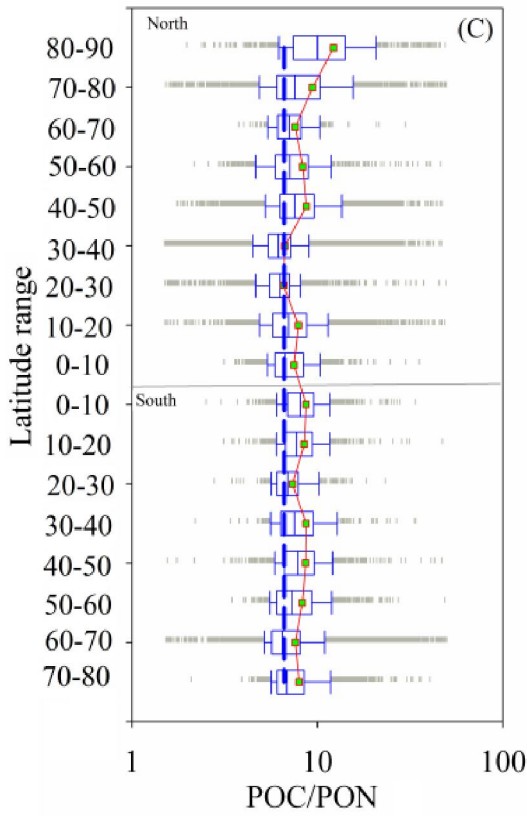

Figure 2 Latitudinal variation of PON, POC and the POC/PON ratio from the compiled statistical results of depth-integrated data for all ocean data. The box plots show the median and 25th and 75th percentiles, with whiskers covering most of the data. The red line with green boxes shows the mean value for each latitudinal range. The bold blue dashed line in (C) is the Redfield value (6.625).

The vertical distributions of POC and PON, and the resulting variations in POC/PON, have a critical impact on

carbon and nitrogen cycles in oceans and should be considered in models of the biogeochemical cycle (Schneider

et al., 2003; 2004; 2005; Babbin et al., 2014). These distributions also indicate that the depth-dependence of POC

and PON in the southern hemisphere (POC: 17.83±2.03 μmol/L ,PON: 2.72±0.29 μmol/L) is stronger than in the

northern hemisphere (POC: 16.79±1.51 μmol/L, PON: 2.49±0.23 μmol/L) (Figure 3A, B). However, the

depth-dependence of POC/PON in the southern hemisphere is weaker than that in the northern hemisphere



(Figure 3C). POC/PON increased significantly, from 6.88±2.3 (0-5 m) to 8.36±6.5 (> 80 m), in the northern

hemisphere but was nearly constant (7.92±0.10) in the southern hemisphere (Table S3). Increases in POC/PON

with depth in the northern and southern hemispheres occurred at rates of 5.2/km and 2.5/km (depth < 200 m),

respectively. These increasing rates are much higher than the 0.2±0.1/km (0 - >5000 m) rate proposed by

Schneider (2004). This may be due to the predominance of nitrogen remineralization in shallow ocean water

(Babbin et al., 2014). The linear slope of POC and PON in the northern hemisphere (7.11±0.36) is larger than in

the southern hemisphere (6.12±0.72) (Table S3) with a global mean value of 6.341±0.856.

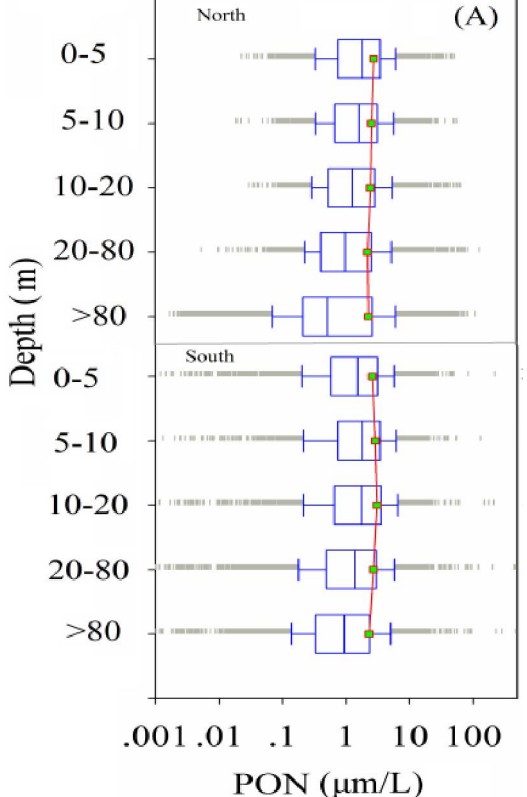
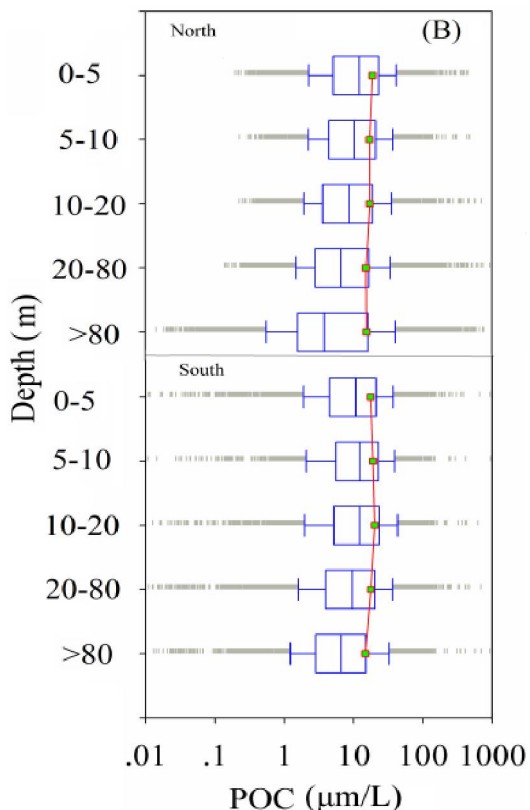

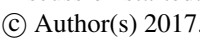



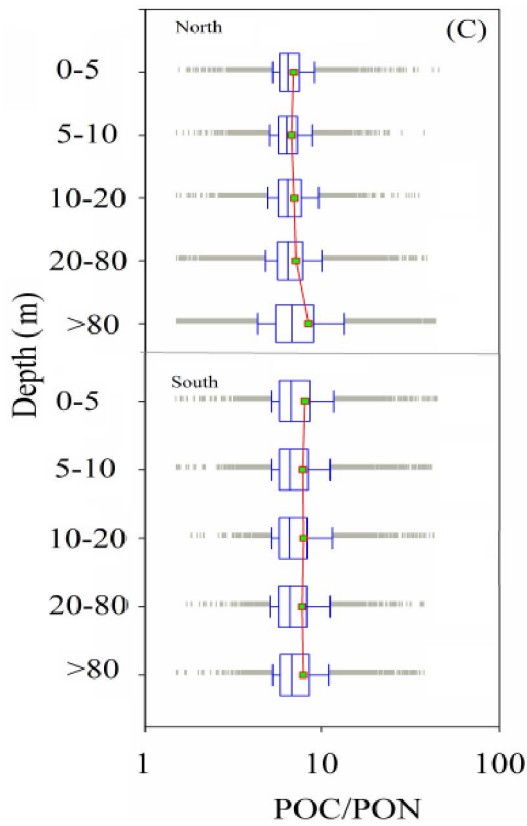

Figure 3 Depth-dependence of POC and PON in the southern hemisphere and depth-dependence of POC/PON in the northern hemisphere. The box plots show the median and 25th and 75th percentiles, with whiskers covering most of the data. The red line with green boxes shows the mean value at each depth. The number of samples for each depth is listed in Table S3.

3.2 Variations in POC, PON and POC/PON with offshore distance

Previous studies indicate that about 0.9 PgC/yr of carbon are discharged from rivers to oceans (Cole et al., 2007); approximately 50% of terrestrial carbon exists in organic form (0.3 PgC/yr of DOC and 0.2 PgC/yr of POC) (Cole et al., 2007; Bianchi, 2011). The enriched terrestrial carbon delivered from rivers causes variations in POC, PON and POC/PON, especially in coastal regions (Martiny et al., 2013b), although a large amount (0.3 PgC/yr) of terrestrial carbon (0.5 PgC/yr) is emitted as $CO_2$ (Cai, W.J., 2011) and most of the remaining terrestrial carbon

sinks as sediment in estuaries.

Variations in POC and PON levels with distance from shore show that there is a significant separation zone (50

km) dividing POC and PON levels into two regions in the northern hemisphere (Figure 4A, B). POC levels close to

land (0 - 50 km) (region of close to shore, 21.90±11.01 μm/L) are nearly two times larger than in regions more

than 50 km from land (region of offshore, 11.65±3.58 μm/L) due to terrestrial influences. The distribution of

PON is similar; PON levels are higher close to shore (3.19±1.46 μm/L, 0 - 50 km) compared to offshore

(1.67±0.44 μm/L, >50 km). Terrestrial impacts on POC and PON levels in the southern hemisphere are relatively

weak (POC, 20.14 ± 5.51 μm/L; PON 3.12 ± 0.87 μm/L) for all distances from shore (Figure 4A, B). In addition,

variation in POC/PON with distance from shore is insignificant in both the northern (7.5 ± 4.6) and southern (7.8 ±

3.8) hemispheres. The POC/PON ratio close to shore (6.89 ± 2.38 in north and 7.59 ± 3.77 in south) is smaller than

in offshore regions (7.59 ± 4.22 in north and 7.90 ± 3.99 in south). Coastal water with relatively high POC and

PON levels has a low POC/PON ratio in the northern and south hemisphere (Table S4). This is inconsistent with

previous studies that show coastal water has a higher POC/PON ratio than offshore water (Sterner et al., 2008;

Kaiser et al., 2014; Watanabe and Kuwae, 2015) due to the discharge of terrestrial organic matter (Hilton et al.,

2015; Cai, Y.H. et al., 2015). The over-consumption of carbon in coastal waters reduces the POC/PON ratio of

terrestrial organic matter. Previous study proposed that more than 0.2 PgC/yr of $CO_2$ is emitted from coastal

waters due to the microbial decomposition of terrestrial organic matter (Cai, W.J. et al., 2011) as well as the

priming effect (Bianchi, 2011). Zooplankton, phytoplankton and high nutrient levels also reduce POC/PON in



coastal waters (Koeve, 2006; Martiny et al., 2013b; Watanabe and Kuwae, 2015; Talmy et al., 2016). The

relatively high POC/PON ratio in offshore water is primary caused by small phytoplankton, which is the

dominant contributor to POC levels at the ocean surface and has a high POC/PON ratio (Richardson and Jackson,

2007; Puigcorbé et al., 2015). The increase of POC/PON with the distance is very significant in the North

hemisphere with the increasing rate of 0.0024/km (POC/PON=0.0024*D+7.1764, $R^2$=0.519), but insignificant in

South hemisphere with rate of 0.0004/km (POC/PON=0.0004*D+7.7346, $R^2$=0.118) (Figure S2).

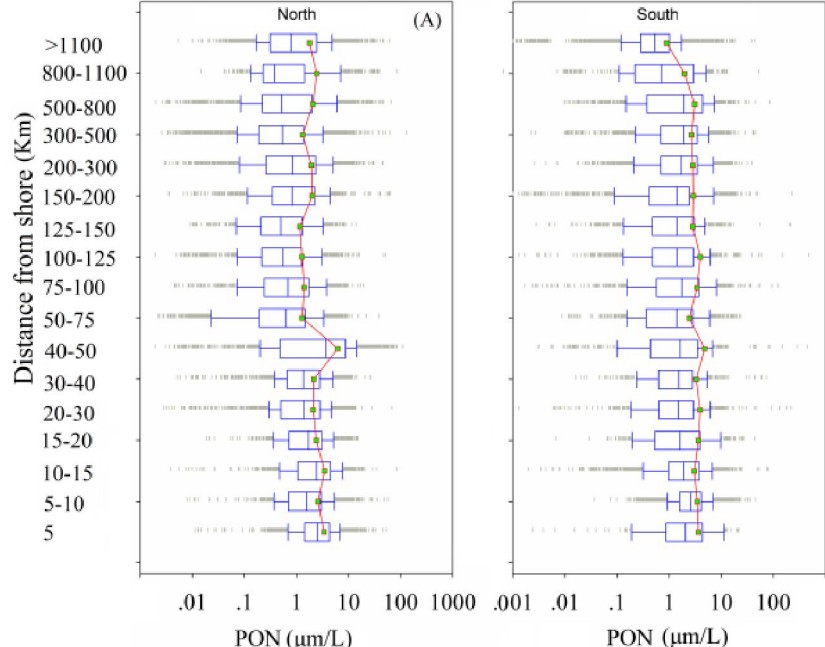





Figure 4 Variation of POC, PON and POC/PON with distance from shore. The number of samples for each buffer is listed in Table S4. The box plots show the median and 25th and 75th percentiles, with whiskers covering most of the data. The red line with green boxes shows the mean value for each range.

3.3 Variability of POC, PON and POC/PON in inland waters

Inland waters play an important role in the global carbon cycle, linking the terrestrial, atmospheric, and oceanic

carbon pools. The lakes and rivers were sorted by the latitude according to the position. POC and PON in inland

waters exhibit a similar trend with that in ocean of north hemisphere, which POC and PON decreased with the

latitude, but greater variability than in oceans due to the strongly dual influences of terrestrial and aquatic organic

matter (Figure 5) (Wilkinson et al., 2013). The POC/PON in inland waters decreased with the latitude. However,

the lake's morphology, trophic state, climate and other influencing factors also cloud regulate latitude-dependent

of POC/PON in inland waters. For example, Kasumigaura Lake is an extremely eutrophic lake, with a mean

(from 1977 to 2013) chlorophyll-a concentration of $67 \pm 44$ µg/L. High productivity in Kasumigaura Lake leads

to relatively high POC and PON levels (POC, 332.76 µm/L and PON, 47.94 µm/L). The trophic state of Lake

Taihu is similar to Kasumigaura Lake (Huang et al., 2015), and the POC and PON levels in Lake Taihu are very

close to those in Kasumigaura Lake. However, the POC/PON ratio in Lake Taihu (4.04) is much smaller than in

Kasumigaura Lake due to over-consumption of organic carbon; large areal lakes, (e.g., Lake Taihu) emit much

more $CO_2$ than small lakes (e.g., Kasumigaura Lake) (Xiao et al., 2014; Hotchkiss et al., 2015). The Great Lakes

(large areal lakes) also have a low POC/PON ratio (5.07). Lakes located in cold-dry climatic zones (McMurdo

Dry Valleys Lakes, Alaskan Lakes, Norwegian Lakes, Lake Baikal, and Hovsgol Lake) tend to have low POC and

PON levels but a high POC/PON ratio (Figure 5). This agrees with previous studies that show that inland waters

maintain high POC/PON ratios due to the strong impact of terrestrial organic matter (Guo et al., 2003; Cai, Y.H.





et al., 2008). The data indicate that the average POC/PON ratios in lakes are approximately 10.6 at the global

scale, 5.67 for deep lakes, 8.16 for shallow lakes, 11.46 for frigid northern lakes and 10.37 for temperate lakes

(Chen et al., 2015). The highest POC/PON ratio appeared in the Ipswich and Parker rivers (28.73). This value is

higher than in previous studies on the Mississippi River (9.8, Trefry et al., 1994; 14.4, Cai Y.H. et al., 2015), the

USA central river (11.22 ± 1.86, Onstad et al., 2000) and the Amazon River (11.6, Moreira-Turcq et al., 2013),

but it is still lower than in northern rivers such as the Chena River (32 ± 12, Guo et al., 2003; 34.33 (Cai, Y.H. et

al., 2008). The relationships of PON and POC for each lake were listed in Table S5.

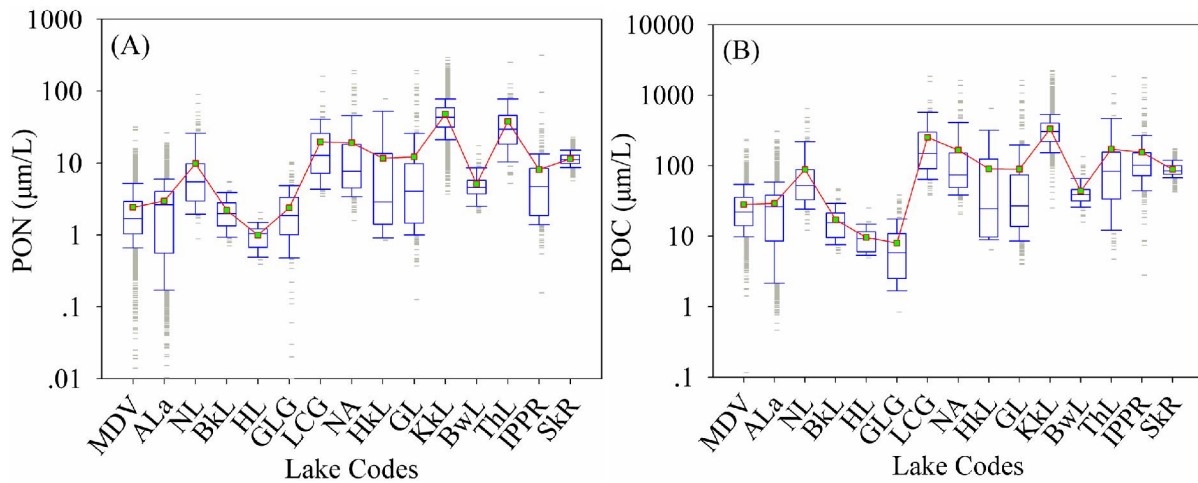





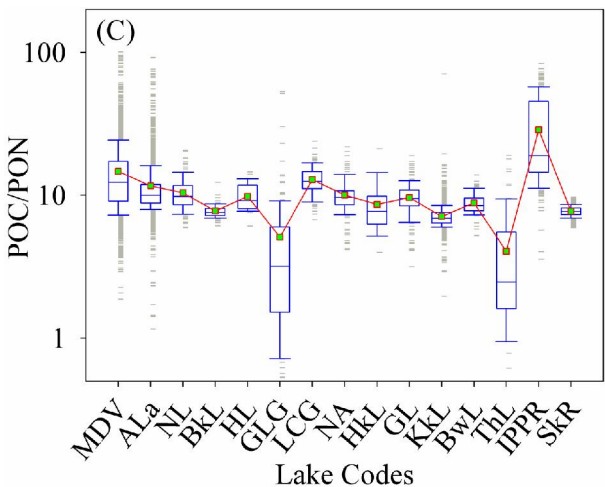

Figure 5 Variation of PON, POC and POC/PON in inland waters. The lakes and rivers are mainly located in the northern hemisphere. Eutrophic, small, large, and shallow lakes are included in the dataset. MDV is McMurdo Dry Valleys Lakes (Antarctica), ALa is Alaskan Lakes(**USA**), NL is Norwegian Lakes (Norway), NA is Northern American Lakes (**USA**), HkL is Hokkaido Lakes (Japan), LCG is Lacustrine Central Group (**USA**), KkL is Lake Kasumigaura (Japan), BwL is Lake Biwa (Japan), HL is Lake Hovsgol (Mongolia), BkL is Lake Baikal (Russia), GL is Green Lake (Canada), ThL is Lake Taihu (China), GLG is Great Lakes Group (**USA**), SkR is Skidaway River (**USA**), IPPR is Ipswich and Parker rivers (**USA**). The box plots show the median and 25th and 75th percentiles, with whiskers covering most of the data. The red line with green boxes shows the mean value of each range. The lake names and their abbreviations were listed in the Table S6.

## 3.4 Drivers of POC, PON and POC/PON variation

### 3.4.1 Terrestrial organic carbon

Land is a huge organic carbon pool and delivers a large amount of POC into oceans via rivers (IPCC, 2013).

Global studies of riverine export of POC have proposed that POC export from land to the oceans is mostly caused by physical erosion (Galy et al., 2015). The storage and distribution of soil organic carbon (Köchy et al., 2015) in the global terrestrial sphere is highly positively correlated to POC and PON levels in the oceans (Figure 6A, B). The linear functions POC = 0.0961*SOC + 3.4355 ($R^2$=0.86) and PON = 0.0103*SOC + 0.5132 ($R^2$=0.83) express the relationship between PON, POC and SOC well, except between of 40 °N ~ 50 °N and 80 °N ~ 90 °N



(marked with an ellipse in Figure 6A, B). PON and POC levels between 40 °N ~ 50 °N are underestimated by the

relationship between PON, POC and SOC due to excess organic matter from phytoplankton (satellite estimation

result chlorophyll-a in ocean color products, http://oceancolor.gsfc.nasa.gov/cgi/l3). Overestimated PON and

POC levels for 80 °N ~ 90 °N are primarily caused by ice on the land and ocean. POC/PON is negatively

correlated to the ratio of land area to coastline length (land/coastline): POC/PON =11.938*(land/coastline)$^{-0.078}$

($R^2$=0.41) (Figure 6C).

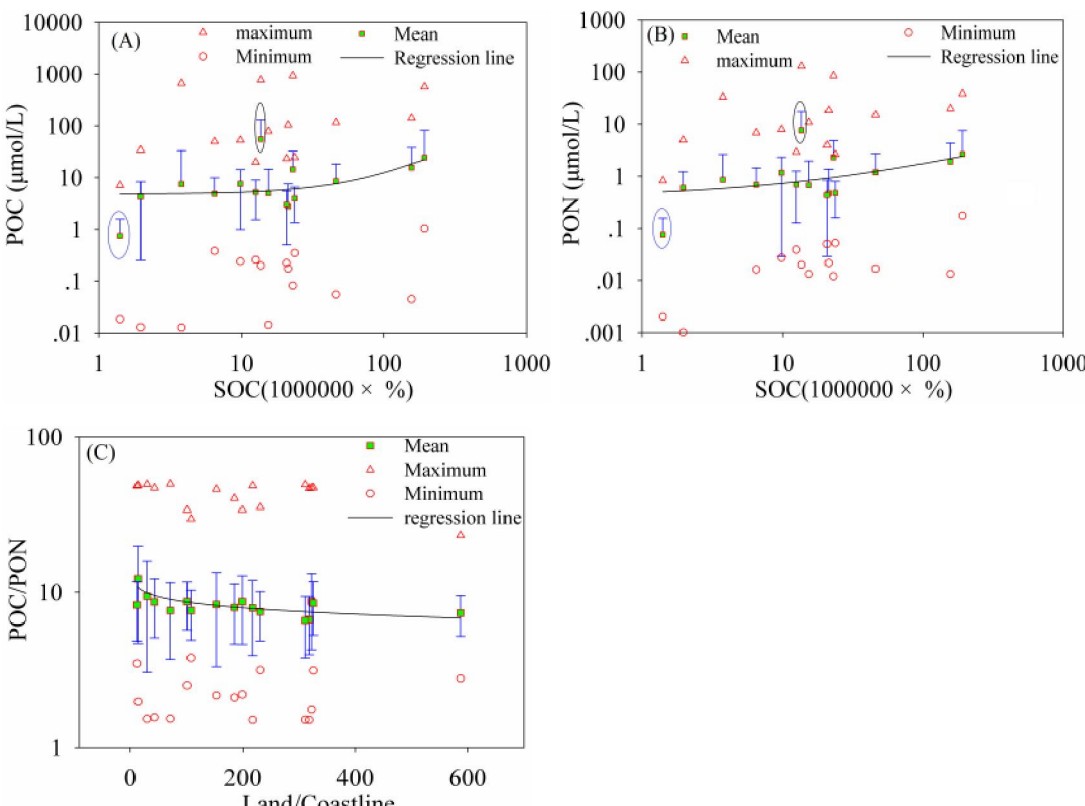

Figure 6 (A) and (B) Relationship between POC, PON and soil organic carbon. (C) The relationship between
POC/PON and the ratio of land area to coastal linear length. The red boxes with green shading are the mean values of
POC, PON and POC/PON in each latitudinal range, the blue line is the standard deviation, and the red triangle and
roundness are the maximum and minimum values of POC, PON and POC/PON in each latitudinal range.



### 3.4.2 Temperature

The temperature dependence of organic carbon production (e.g., primary production, released from permafrost and soil erosion) and consumption (e.g., mineralization, respiration and methane emission) increases the influence of temperature on aquatic ecosystems and reflects the importance of temperature in the carbon cycle (Gudasz et al., 2010; Padfield et al., 2015; Yvon-Durocher et al., 2011a; 2011b; 2012; 2014; 2015a; 2015b; Zona et al., 2016). POC and PON levels are highly positively correlated to temperature in the northern hemisphere with the relationships PON=0.142*T+0.260 ($R^2$=0.74) and POC=0.788*T+3.340 ($R^2$=0.74). However, the effect of temperature on POC and PON levels in the southern hemisphere is not very significant, with correlation coefficients ($r$) of -0.11 and -0.08, respectively. The influence of temperature on POC and PON levels at a global scale is not homogeneous (Figure 7A, B). The increased sensitivity of POC and PON to temperature in the northern hemisphere may be caused by relatively large amounts of nutrients and a large land area when compared to the southern hemisphere. POC/PON is highly negatively correlated to temperature in the northern hemisphere, with the relationship POC/PON=11.88*$T^{-0.190}$ ($R^2$=0.81) (not including samples with subzero temperature). Phytoplankton and microzooplankton growing in low temperatures (subzero) may regulate POC/PON, keeping the value low (Crawford et al., 2015; Talmy et al., 2016), and nitrogen ($NO_3^-$ and $NH_4^+$) uptake and light may also play a role (Yun et al., 2012). The impact of temperature on POC/PON in the southern hemisphere ($r$=-0.31) is relatively low when compared to the northern hemisphere (Figure 7C). This may indicate that the mineralization of organic carbon occurs at a much higher rate than organic nitrogen with increasing temperature or that terrestrial organic carbon, which has a high POC/PON ratio, is more efficiently kept than phytoplankton- and



microzooplankton-derived organic carbon, which has a low POC/PON ratio, with increasing temperature

(Sharma, et al., 2015; Porcal et al., 2015; Watanabe and Kuwae, 2015; Crawford et al., 2015; Talmy et al., 2016).

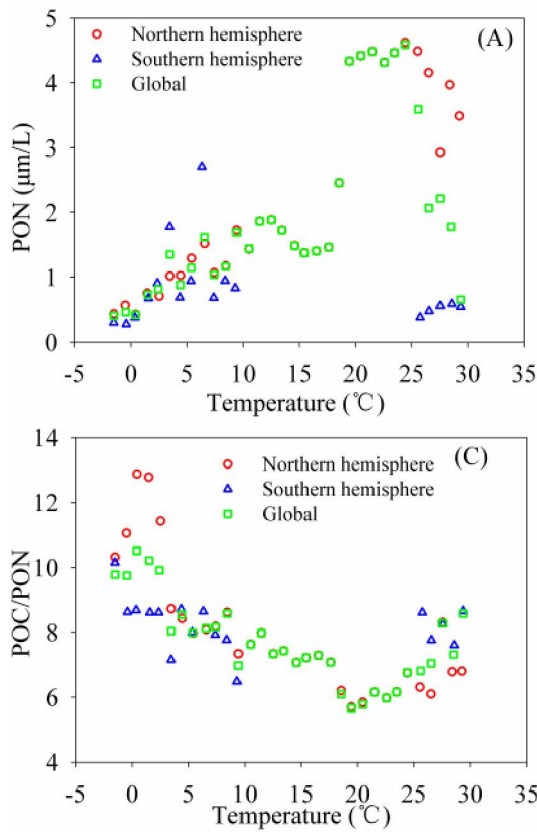

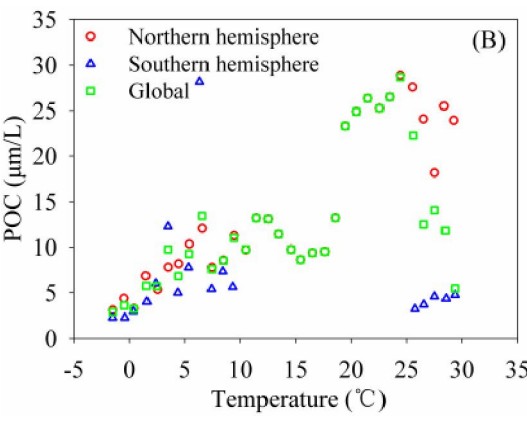

Figure 7 Relationships between POC, PON, and POC/PON and temperature (T). POC, PON and POC/PON are highly correlated to temperature in the northern hemisphere, with relationships of PON=0.142*T+0.260 ($R^2$=0.74), POC=0.788*T+3.340 ($R^2$=0.74) and POC/PON=11.88*T$^{-0.190}$ ($R^2$=0.81), respectively. There is almost no correlation in the southern hemisphere, with correlation coefficients of -0.11, -0.08 and -0.31 for PON, POC and POC/PON, respectively. The relationships between POC, PON, and POC/PON and T at a global scale are PON=0.093*T+0.697 ($R^2$=0.42), POC=0.507*T+5.630 ($R^2$=0.41) and POC/PON=10.02*T$^{-0.122}$ ($R^2$=0.57).

### 3.4.3 Productivity and migration

Phytoplankton is an agent of the biological pump, which sequesters carbon from the atmosphere to the deep sea

(Koeve, 2006); thus, it influences the global carbon cycle and the climate system (Lam et al., 2011). Studies have

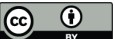

proposed general relationships between POC and chlorophyll-a concentration ($C_{Chl-a}$) to describe the dominant

effect of phytoplankton on the POC reservoir (Peña et al., 1991; Legendre and Michaud, 1999; Lefevre et al.,

2003; Wang et al., 2011; Wang et al., 2013). However, the relationship between POC and $C_{Chl-a}$ can't accurately

explain POC variation at the global scale due to high variation in POC/$C_{Chl-a}$ (Arrigo et al., 2003; Sathyendranath

et al., 2009). POC levels are highly positively correlated to $C_{Chl-a}$ for oceans, lakes, eutrophic lakes, rivers and

coastal waters (Figure 8A). The best fit function for the relationship between POC and $C_{Chl-a}$ varies with water

type. A linear function is best for lakes ($R^2$=0.64) and coastal waters ($R^2$=0.82), and a power function is best for

oceans ($R^2$=0.68) and eutrophic lakes ($R^2$=0.77). Both linear and power functions can be used in rivers (linear,

$R^2$=0.77; power, $R^2$=0.77). This is partly consistent with previous studies on ocean water, where POC co-varied

with $C_{Chl-a}$ via a power function (Sathyendranath et al., 2009; Wang et al., 2011). However, the power exponent

for the global ocean (0.581) is slightly higher than that of Wang (0.5402, 2011). The highest power exponent

(0.645) is for eutrophic lakes, and the lowest (0.434) is for rivers. The slope of the linear fit function in lakes is

much higher than in coastal waters. The best-fit power function for POC and $C_{Chl-a}$ in oceans, eutrophic lakes and

rivers demonstrates that phytoplankton carbon sequestration efficiency reduces with increasing chlorophyll-a in

these water types. Consequently, carbon sequestration efficiency in eutrophic lakes, following a power function,

is much higher than in oceans and rivers, and in lakes, following a linear function, it is higher than in coastal

waters. Thus, the regulation of lake water requires more attention, as it significantly affects the global carbon

cycle.


Total suspended particulate matter transported from the continental biosphere significantly affects POC levels in

the water body, in addition to producing phytoplankton (Galy et al., 2015). The relationship between POC and

suspended particulate matter concentration ($C_{\text{TSM}}$) (Figure 8B) is very similar to studies that show that POC is highly

positively correlated to suspended particulate matter (Ni et al., 2008; Cetinić et al., 2012; Woźniak et al., 2016;

Yang et al., 2016). The linear relationship between POC (µm/L) and $C_{\text{TSM}}$ (mg/L) is shown in Figure 7B. The

power relationship between POC (mg/L) and $C_{\text{TSM}}$ (mg/L) at the global scale (POC=0.2641*$C_{\text{TSM}}^{0.8466}$, $R^2$=0.81,

n= 5306) is close to the same relationships in the Baltic Sea (POC = 0.317*$C_{\text{TSM}}^{0.969}$, $R^2$=0.86, Woźniak et al., 2016)

and surface water in the US (POC = 0.2992*$C_{\text{TSM}}^{0.3321}$, $R^2$=0.593, Yang et al., 2016). However, this relationship

differs slightly from the one presented by Galy (2015, Figure 1–3), in which the global flux of terrestrial POC to

oceans is composed of biospheric (80%) and petrogenic (20%) POC, with the relationship $POC_{\text{exp}}$ =

0.0524*$C_{\text{sed}}^{0.665}$. This indicates that suspended particulate matter includes large amounts of organic carbon in

addition to terrestrial organic carbon due to primary productivity and the subsequent zooplankton in the food

chain.

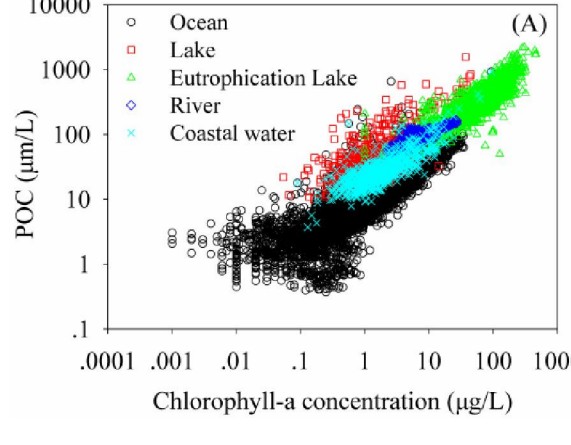
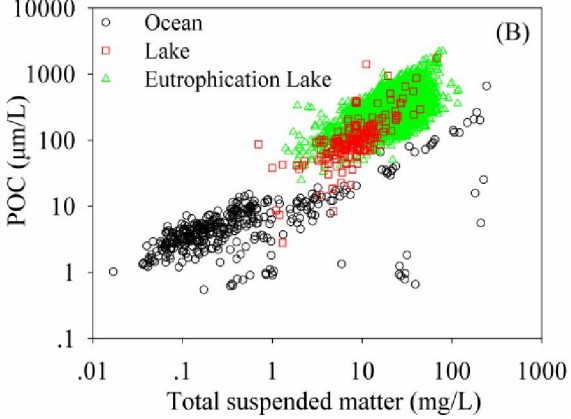





Figure 8 (A) Comparison of the relationships between POC and chlorophyll-a concentration ($C_{\text{Chl-a}}$) for oceans, lakes, eutrophic lakes, rivers and coastal waters. (B) Comparison of the relationship between POC and total suspended matter concentration ($C_{\text{TSM}}$) for oceans, lakes and eutrophic lakes. The relationships between POC and $C_{\text{Chl-a}}$ are POC=4.429*$C_{\text{Chl-a}}$+6.531 ($R^2$=0.51, N=5462); POC=16.420*$C_{\text{Chl-a}}$+17.855 ($R^2$=0.64, N=984); POC=3.513*$C_{\text{Chl-a}}$+96.528 ($R^2$=0.72, N=4656); POC=4.458*$C_{\text{Chl-a}}$+55.931 ($R^2$=0.77, N=936); and POC=7.357*$C_{\text{Chl-a}}$+12.349 ($R^2$=0.82, N=692) for oceans, lakes, eutrophic lakes, rivers, and coastal waters, respectively. The relationships between POC and $C_{\text{TSM}}$ are POC=1.045*$C_{\text{TSM}}$+5.198 ($R^2$=0.53, N=432); POC=15.932 *$C_{\text{TSM}}$-25.645 ($R^2$=0.67, N=191); and POC=7.984*$C_{\text{TSM}}$+149.950 ($R^2$=0.27, N=4683) for oceans, lakes, and eutrophic lakes, respectively.

### 3.4.3 Dissolved organic carbon and nitrogen

DOC, which is present in much higher concentrations than POC (Figure 8A and Figure 9A), quantitatively represents the most important carbon pool (Emerson and Hedges, 2008). DOC is a complex mix of organic compounds from both autochthonous and allochthonous sources that primarily originate from aquatic organisms and runoff, respectively (Doval et al., 2016; Kuliński et al., 2016). DON is highly positively correlated to DOC, with a best-fit function of DOC=17.825*$DON^{1.019}$ ($R^2$=0.58, n=995) at the global scale. The linear regression model (Figure 9A) shows that the slope of the linear function is smaller than in previous regional studies (13.3 ± 0.8, Doval et al., 1999; 20.5 ± 3, Aminot and Kérouel, 2004) and is also smaller than DOC sequestered in the deep sea (17.38) via the microbial carbon pump (Jiao et al., 2010). The DOC/DON ratio in lakes (40.43 ± 34.56) and rivers (29.35 ± 34.93) is much higher than in oceans (12.86 ± 4.88) and coastal waters (13.15 ± 4.95) (Figure 9B); thus, inland water holds much more DOC, which may result in the high emission of CO2 in inland waters (Raymond et al., 2013; Hotchkiss et al., 2015). The correlation between POC and DOC is weak for individual water types, but is strong for the whole data of lake, ocean and coastal water. The regression function (gray line in the



Figure 9 C) between POC and DOC is DOC = 0.315 * POC + 64.88 ($R^2$=0.58 n=570), except the data of river.

High DOC/POC and DOC/DON ratios indicate that organic carbon is mostly stored in dissolved form. DOC and

DON regulate the organic carbon and nitrogen equilibrium system with the POC and PON, besides the interaction

with the inorganic carbon and nitrogen (such as $CO2$, nutrients, $N_2$ and $NO_x$).

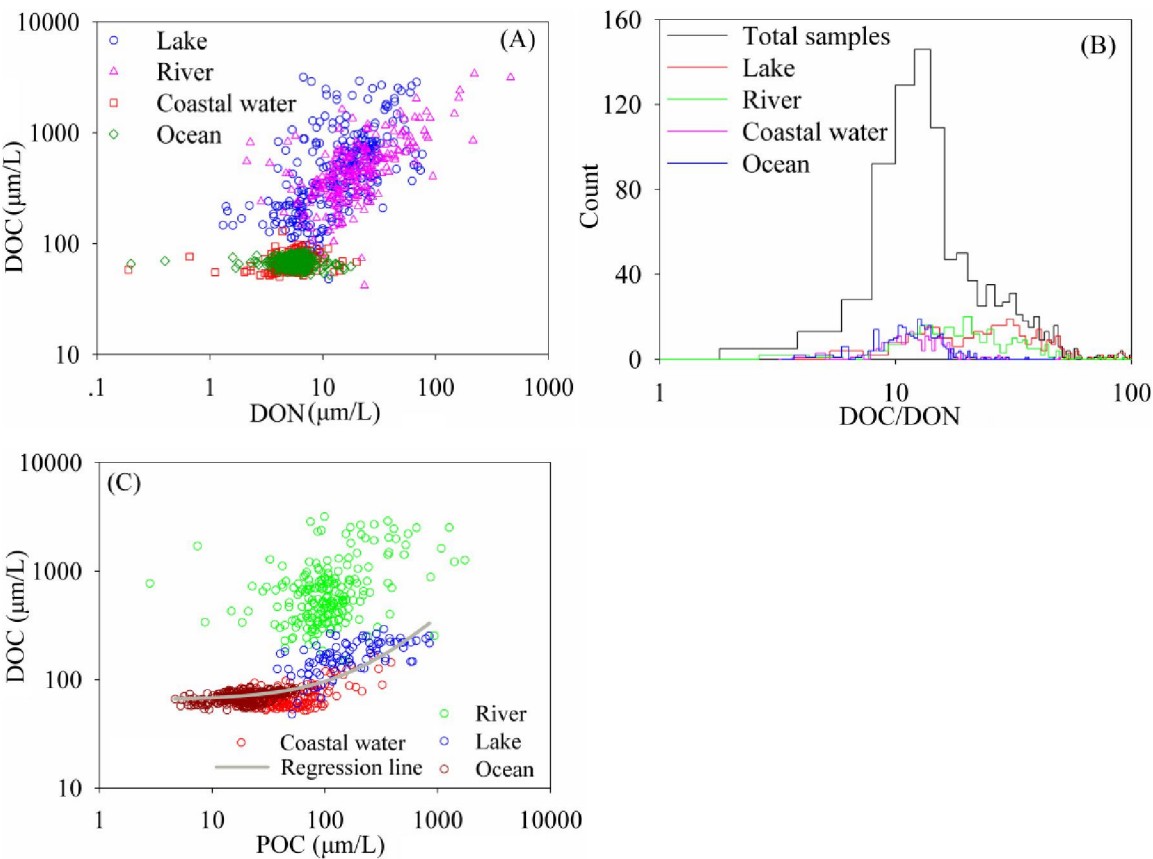

Figure 9 (A) Comparison of the relationships between DOC and DON for oceans, lakes, rivers, and coastal waters. The relationships between DOC and DON are DOC = 11.738*DON + 165.41 ($R^2$=0.35, n=995); DOC = 19.037*DON + 273.59 ($R^2$=0.20, 288); and DOC = 11.932*DON + 234.36 ($R^2$=0.61, n=255) for all samples, lakes and rivers, respectively. The correlation coefficients for DOC and DON are r=0.44, 0.78, 0.15 and -0.15 for lakes, rivers, coastal waters and oceans, respectively. (B) DOC/DON for oceans, lakes, rivers and coastal waters. The DOC/DON ratios are
24.10 ± 24.57, 40.43 ± 34.56, 29.35 ± 34.93, 13.15 ± 4.95 and 12.86 ± 4.88 for all samples, lakes, rivers, coastal waters and oceans, respectively. The corresponding POC/PON ratios are 9.65 ± 16.73, 7.71 ±0.61, 7.07 ± 2.44 and 6.24 ± 1.20 for lakes, rivers, coastal waters and oceans, respectively. (C) Relationships of POC and DOC for different



water types. The correlation between POC and DOC is weak for individual water types, but is strong for the whole data of lake, ocean and coastal water. The regression function (gray line in the figure) between POC and DOC is

DOC = 0.315 * POC + 64.88 ($R^2$=0.58, n=570), except the data of river.

Analysis of global temporal and spatial variation in POC, PON and POC/PON and the analysis of drivers that

influence POC, PON and POC/PON distribution is the basis of biogeochemical implication. The simple mean

value of POC/PON at the global scale (7.54±3.82) is higher than the Redfield ratio (6.63), but the linear

regression slope (including intercept, 6.17; excluding intercept, 6.23) for all ocean data is much lower than the

simple mean value of POC/PON and the Redfield ratio. The linear regression slopes between POC and PON in

the northern hemisphere (including intercept, 7.06 and excluding intercept, 7.00) are much higher than in the

southern hemisphere (including intercept, 5.97; excluding intercept, 6.03). Variations in POC, PON and

POC/PON in inland waters requires further attention due to the importance of inland waters in global carbon and

nitrogen cycles and the indeterminacy of the relationship between carbon and nitrogen. Land and soil organic

carbon distribution and offshore distance were appeared to be controlled factors to the variation of POC, PON

and POC/PON at a global scale besides the temperature and productivity.

**Acknowledgments**

This study was supported by the National Natural Science Foundation of China (Grant Nos. 41571324, 41673108 and 41503075), a project funded by the Priority Academic Program Development of Jiangsu Higher Education
Institutions, and the Innovation driven development capacity building project of Guangdong Academy of Sciences (2017GDASCX-0801). Support from A-Xing Zhu through the Vilas Associate Award, the Hammel Faculty Fellow Award, the Manasse Chair Professorship from the University of Wisconsin-Madison, and the "One-Thousand Talents" Program of China is greatly appreciated. We are grateful to Nick Kleeman for the language editing.



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
