# Peer review of "Variation pattern of particulate organic carbon and nitrogen in oceans and inland waters"

_Biogeosciences, 2017_

## Referee Comment (RC1) · Anonymous Referee #1 · 9 Apr 2017

General Comments:

The authors have performed a review of variability in particulate organic carbon and nitrogen ratios in the world's oceans and inland waters. While the authors include a large amount of data for the ocean that is more or less globally representative, they consider 2 small temperate rivers and 7 different lakes, all located in the Northern hemisphere. There doesn't appear to have been any effort to incorporate data from the vast body of literature, rather data was only downloaded from websites with data readily available. For such a review to be meaningful, the authors need to spend considerable time mining the literature to collect a representative dataset. Not including large rivers in a global dataset is a massive oversight.

Currently I do not see any value in this review considering the massive gaps in the

data that was considered. The authors perhaps have a decent starting point with the assembled ocean datasets, but need to spend considerable time compiling the inland water data before any meaningful conclusions can be made. I have recommended some references to read through below to broaden perspectives on inland water biogeochemical cycling that will perhaps inspire a more in depth analysis.

Specific Comments:

Line 18: It would be perhaps more interesting to first list the difference between inland waters, oceans, and estuarine C/N averages, rather than just a global average. . .this comment was made before realizing how sparse the inland water dataset is.

Line 20: C/N variability in inland waters was attributed to "lake geomorphology, trophic state, and climate." This is a vast oversimplification, which is reflective on the manuscript in general. Rivers are not even mentioned, which are highly dynamic. For example, C/N ratios (either dissolved or particulate) can vary by several times over the course of a few hours in rivers/streams in response to rainfall. This concept is discussed in the following manuscript and the references therein and should be considered for further discussion in the manuscript:

Ward, N.D., Keil, R.G., Richey, J.E. (2012) Temporal variation in river nutrient and dissolved lignin phenol concentrations and the impact of storm events on nutrient loading to Hood Canal, Washington, USA. Biogeochemistry. 111 (1-3), 629-645

The above comment was made prior to realizing the inland water dataset only included lakes and 2 rivers. Now this focus makes sense. . .

Line 30-35: There are much more recent syntheses of global inland water $CO_2$ budgets that should be mentioned if this is going to be the focal point of the first paragraph. For example, see the following refs. Raymond et al. (2013) increased the outgassing component to 2.1 Pg C yr. Sawakuchi et al., (2017) noted, that a large fraction of the surface area of the world's inland waters aren't accounted for. . .adding the complete

surface area just of the Amazon River increases the global budget to 2.9 Pg C yr-1. This progression and factors that are still missing from global budgets were discussed in the review paper by Ward et al. (2017):

Raymond, P.A., Hartmann, J., Lauerwald, R., Sobek, S., McDonald, C., Hoover, M., et al. (2013). Global carbon dioxide emissions from inland waters. Nature. 503(7476), 355-359

Sawakuchi, H.O., Neu, V., Ward, N.D., Barros, M.L.C., Valerio, A.M., Gagne-Maynard, W., Cunha, A.C., Less, D.F., Diniz, J.E., Brito, D.C., Krusche, A.V., Richey, J.E. (2017) Carbon dioxide emissions along the lower Amazon River. Frontiers in Marine Science. 4 (76)

Ward, N.D., Bianchi, T.S., Medeiros, P.M., Seidel, M., Richey, J.E., Keil, R.G., Sawakuchi, H.O. (2017) Where carbon goes when water flows: Carbon cycling across the aquatic continuum. Frontiers in Marine Science. 4 (7)

Line 50: See previous comment on Line 20. The factors controlling C/N in terrestrial environments and inland waters are grossly oversimplified. C/N in inland waters is not only a result of OM origin. Molecules are selectively leached from soils during mobilization into streams (or even the flow paths that come before this such as throughfall, stemflow, etc). Molecules are also selectively degraded and sorbed/desorbed during transport, influencing composition. The review paper mentioned above is a good place to start for honing the conceptualization and discussion of inland waters.

Line 80: After reviewing the list of data used, it is not surprising to see the lack of inland water discussion. There is one river dataset listed as far as I can tell—the Ipswich and Parker rivers, 2 fairly small temperate rivers. The other inland water datasets are from 7 lakes. While the ocean dataset seems to be decently large, the attempt at a "global synthesis" of inland waters made here is non-existent. Where is the Amazon River, which makes up 20% of the freshwater flow to the ocean? How about the Congo River, the Ganges-Brahmaputra River, the Changjiang River, and all of the world's large

rivers? Not to mention streams from different settings.

I would recommend reading the following review from the 1980's that did a more comprehensive job than done here:

Meybeck, M. (1982). Carbon, nitrogen, and phosphorus transport by world rivers. Am. J. Sci. 282(4), 401-450

For this present study to be meaningful, the authors need to include the majority of robust datasets currently available in the literature. It appears the authors only used data that could be readily downloaded from websites, rather than making a true effort to mine the literature. They have ignored the entire body of inland water literature.

---

## Referee Comment (RC2) · Anonymous Referee #2 · 21 Apr 2017

This paper expanded the global marine dataset on POC and PON, including extending the range northward a few degrees of latitude, and produced many new insights or conclusions compared to previously published studies. It's also good to see freshwater data included, and got some evidence of variability in different lake data. Such as, the finding of high C:N at high northern latitudes (ms. Fig. 2) is as far as I know novel and more or less inverts the temperature-based conclusions of Martiny et al. (who showed C:N increasing with temperature). The ms. figure 7C is quite different from what Martiny et al. (2013) showed in their figure 4. These new insights or conclusions compared to previously published studies suggest that there still are some critical things we need to know to deepen our understanding of global patterns in linkages of C and N. The authors have performed a great service in assembling these data and this is important to extend current knowledge to a wider range of geography. Considering new scientific

findings in this paper, i recommend a potential acceptance in this journal, but necessary revisions are needed. I have the following specific comments and suggestions: 1) Title – What is meant by "variation pattern?" Suggest a more descriptive title would be something like "Global patterns in particulate and dissolved organic carbon and nitrogen in the global ocean and inland waters." 2) Figures – All of them are too small, which made it really hard to see what was going on with the data. Suggest converting each on to landscape orientation and then filling the entire page with it, or submitted each figure respectively. 3) The Abstract is adequate. but the means of (12.2±7.5) should be noted, mean value ± error or standard deviation? 4) The Introduction is okay but not very inspiring. 5) I believe the analysis of distance to land (Fig. 4) is by far the most extensive one yet. The detailed analysis method should be introduced in section 2.2, although '3) Offshore distance ranges...........' was mentioned. 6) The analysis concerning soil carbon and nitrogen is novel. However, there was no mention in the Methods as to where these soil data come from or how they were matched to the marine data.

There are some really intriguing patterns here that depart from previous work and which are based on what I believe to be the most comprehensive dataset yet assembled on these parameters although some imperfections should be polished. This dataset has some interesting patterns that will help us move stoichiometry forward.
* * *

---

## Author Response (AR1)

Dear Prof. Gerhard Herndl

First, we very appreciate the useful comments from two anonymous reviewers and these comments are very meaningful to improve the quality of this paper and our further study.

Our paper is a research paper about the stoichiometry of particulate organic carbon (POC) and particulate organic nitrogen (PON) based on expanded the global marine dataset and additional large number of inland water data. The first anonymous reviewer focuses on stoichiometry of POC and PON in the global ocean water and gave us a positive evaluation. However, the anonymous reviewer is much interesting in the review of POC, PON and carbon cycle (such as $CO_2$ emission from inland water) in inland water and point out the data insufficient of POC and PON in inland water of our study. This is very useful and meaningful for our further study and gets an ambitious objective for global study of carbon cycle, mapped by reviewer.

Compared to the stoichiometry of POC and PON in inland water, the stoichiometry of POC and PON in the global ocean water is much stable. We also got some new findings different from previous studies based on the expanded the global marine dataset. In personal mind, we did not necessary to collect all the inland water data for the analyses of variation of POC/PON and inland water data worked as subsidiary in this study. The inland data of POC and PON includes 11875 samples in 253 lakes (small lakes in similar region were not listed separately in the supporting information). The analysis is validated if the representativeness of analyses dataset for the inland water can satisfy the variation range of such 'global synthesis'. Of course, the representativeness will be increased with much more data was utilized, and we should pay more attention to the complete collection of inland water data if we want to estimate the discharge flux to the ocean from rivers in further study.

Nonetheless, we added much more data of POC and PON in the world wide river in the revision edition and expected to improve the acknowledge of stoichiometry in the inland water according to the reviewer's comment.

We marked some major revisions in the context. the revised supporting information also was submitted.

**Anonymous Referee #1**

General Comments:

1)  The authors have performed a review of variability in particulate organic carbon and nitrogen ratios in the world's oceans and inland waters. While the authors include a large amount of data for the ocean that is more or less globally representative, they consider 2 small temperate rivers and 7 different lakes, all located in the Northern hemisphere. There doesn't appear to have been any effort to incorporate data from the vast body of literature, rather data was only downloaded from websites with data readily available. For such a review to be meaningful, the authors need to spend considerable time mining the literature to collect a representative dataset. Not including large rivers in a global dataset is a massive oversight. Currently I do not see any value in this review considering the massive gaps in the data that was considered. The authors perhaps have a decent starting point with the assembled ocean datasets, but need to spend considerable time compiling the inland water data before any meaningful conclusions can be made. I have recommended some references to read through below to broaden perspectives on inland water bio-geochemical cycling that will perhaps inspire a more in depth analysis.

Response: We very appreciate the useful comments proposed by reviewer. We responded each comment carefully and did corresponding revision in the context of MS.

In fact, the purpose of this study is not review the variation of POC and PON in ocean and inland water. We compiled large of ocean (63208 samples) and inland (11875 samples, 253 lakes) data of POC and PON not only from online database but also previously studies. The lake dataset listed in the supporting information includes many lake groups. The lake group contains many lakes, not just a single lake. We can open the online link if we are interested in it. These data were used to reexamine the variation pattern and relationship of POC and PON and also compared with previous study (Martiny et al., 2013a and 2013b). We also compared the relationship of POC and PON between ocean and inland water to help us much more comprehensive understanding the variation pattern of POC and PON, although the samples in the inland water is relatively small. Some new variation pattern of POC and PON was revealed via the expanded global marine data and some inland water data.

Martiny, A.C., Pham, C.T.A., Primeau, F.W., Vrugt, J.A., Moore, J.K., Levin, S.A., Lomas, M.W., 2013a. Strong latitudinal patterns in the elemental ratios of marine plankton and organic matter. Nature geoscience 6:279-283.

Martiny, A.C., Vrugt, J.A., Primeau, F.W., Lomas, M.W., 2013b. Regional variation in the particulate organic carbon to nitrogen ratio in the surface ocean. Global Biogeochem. Cycles, 27, 723–731.

Expert (reviewer) expect a thorough study and the summarization and analyses of the global dataset of POC and PON in ocean and inland water, and broaden the POC and PON to the $CO_2$ in global inland water, and also proposed to separate the inland water into lake and river. This is very useful and meaningful for our further study and gets an ambitious objective, mapped by reviewer.

In personal mind, we did not necessary to collect all the inland water data for the analyses of variation of POC/PON and worked as subsidiary in this study. The analysis is validated if the representativeness of analyses dataset for the inland water can satisfy the variation range of such 'global synthesis'. Of course, the representativeness will be increased with much more data was utilized, and we should pay more attention to the complete collection of inland water data if we want to estimate the discharge flux to the ocean from rivers in further study.

**Nonetheless, We added much more data of POC and PON in the world wide river in the revision edition and expected to improve the acknowledge of stoichiometry in the inland water according to the reviewer's comment.**

'Rivers not only bridge the carbon and nitrogen elemental cycles in the land and ocean through the transmission of organic matters, but also conduct the emission of CO2 in the inland water (Raymond et al., 2013). The POC and PON in the river are relatively higher than that in the ocean and lake, especially in the big and high turbid rivers (such as Yanagtze River, Amazon River in figure 6). The POC and PON in Yanagtze River (the highest POC and PON river, 4154.6 ± 3109.6 μm/L and 290.7 ± 180.5μm/L) are proximate 100 and 80 times bigger than in Fraser River (the lowest POC and PON river in this study, 39.7 ± 54.9 μm/L, 3.7 ± 4.3 μm/L), indicating the hugely spatial and temporal variation of POC and PON in river system. It also could manifest that big rivers with high POC and PON may discharge much more POC and PON into ocean (globally annual fluxe of POC is 216 Tg, Voss, 2009) accompanied by high runoff. However, the variation of POC/PON (variable coefficient, 0.47) in river waters is much small than POC (variable coefficient, 2.03) and PON (variable coefficient, 1.81) in river water. The highest POC/PON ratio appeared in the Ipswich and Parker rivers (IPPR, 28.73 in

figure 6C). This value is higher than in previous studies on the Mississippi River (9.74 ± 0.70, this study; 9.7 Trefry et al., 1994; 14.4, Cai Y.H. et al., 2015), the USA central river (11.22 ± 1.86, Onstad et al., 2000) and the Amazon River (10.8 ± 3.3, this study; 11.6, Moreira-Turcq et al., 2013), but it is still lower than in northern rivers such as the Chena River (32 ± 12, Guo et al., 2003; 34.33 (Cai, Y.H. et al., 2008). The lowest POC/PON ratio appeared in the Pearl River (PR, 6.02 ± 1.91), closing to the 5.67 for deep lakes (Chen et al., 2015). The latitude-dependent of POC, PON and POC/PON were not evaluated due to that the samples of each river were not follow the latitude-distribution. The relationships of PON and POC for each river were listed in Table S6.

[Figure]

Figure 6 Variation of PON, POC and POC/PON in river waters. PR is Pearl River (China), YZR is Yanagtze River (China), PDSR is paraiba do sul River (Brazil), AMR is Amazon River (Brazil), FLR is Fly River (Papua New Guinea), FRR is Fraser River (Canada), YKR isYukon River (USA), MISR is Mississippi River (USA),   RUR is Russian rivers (Russian), PIR is Ping River (Thailand), USR is Union and Skokomish River (USA), ORR is Orinoco river(Venezuela), DMR is Mandovi river (India), SkR is Skidaway River (USA), IPPR is Ipswich and Parker rivers (USA).'

Specific Comments:
2)   Line 18: It would be perhaps more interesting to first list the difference between inland waters, oceans, and estuarine C/N averages, rather than just a global average..... this comment was made before realizing how sparse the inland water dataset is.

Response: Actually, we collect large data of POC and PON from different lakes. We have more than11875 couples of POC and PON in the 253 lakes and 15 rivers. We listed the lake dataset in the supporting information, including 13 lakes' and 15 rivers' database.

We shown the variation of POC, PON and POC/PON in ocean and inland water separately primary due to facilitate comparison of our study to previous studies and reveal some new variation pattern of POC, PON and POC/PON in ocean. Meanwhile, the variation of POC, PON and POC/PON in ocean and coastal water has been studied in previous studies. The mean value of POC/PON in ocean is obviously lower than that in inland waters.

The difference of POC, PON and POC/PON in ocean and coastal water was more clearly and detailed in the section 3.2 'variations in POC, PON and POC/PON with offshore distance'. The box chart for different data categories (Latitude-, depth-dependent and so on), included much more statistic information than simple mean value, were used to describe the distribution and variation of POC, PON and POC/PON.

3)   Line 20: C/N variability in inland waters was attributed to "lake geomorphology, trophic state,

and climate." This is a vast oversimplification, which is reflective on the manuscript in general. Rivers are not even mentioned, which are highly dynamic. For example, C/N ratios (either dissolved or particulate) can vary by several times over the course of a few hours in rivers/streams in response to rainfall. This concept is discussed in the following manuscript and the references therein and should be con-sidered for further discussion in the manuscript:

Ward, N.D., Keil, R.G., Richey, J.E. (2012) Temporal variation in river nutrient and dissolved lignin phenol concentrations and the impact of storm events on nutrient loading to Hood Canal, Washington, USA. Biogeochemistry. 111 (1-3), 629-645

The above comment was made prior to realizing the inland water dataset only included lakes and 2 rivers. Now this focus makes sense.....

Response: Yes, this description is an incomplete picture of impact factors to the variation of POC, PON and POC/PON in inland water. We knew that the POC, PON and POC/PON varied with many impact factors. Just as expert have said, the POC, PON and POC/PON can vary by several times due to the variation of streams in rivers, or the sediment resuspension in shallow lakes. However, on the one hand it is hard to get the time series data of POC and PON for worldwide rivers; on the other hand this is not our issue in this paper. Spatial variation with depth, latitude, and offshore distance is prior in our study relative to the temporal variation. Sometimes, variation trend of POC, PON and POC/PON (such variation range) in spatial scale may similar to that in temporal scale. We recognized the temporal variation in river and ocean. We cited the reference (Ward et al., 2012) and added the discussion in the revised context. The influencing factors to the variation of POC/PON are very complex (refer to the Response of comment 5). These impact factors were compacted in the abstract. We showed some detailed information in the corresponding context.

4) Line 30-35: There are much more recent syntheses of global inland water CO2 budgets that should be mentioned if this is going to be the focal point of the first paragraph. For example, see the following refs. Raymond et al. (2013) increased the outgassing component to 2.1 Pg C yr. Sawakuchi et al., (2017) noted, that a large fraction of the surface area of the world's inland waters aren't accounted for..... adding the complete surface area just of the Amazon River increases the global budget to 2.9 Pg C yr-1. This progression and factors that are still missing from global budgets were discussed in the review paper by Ward et al. (2017):

Raymond, P.A., Hartmann, J., Lauerwald, R., Sobek, S., McDonald, C., Hoover, M., et al. (2013). Global carbon dioxide emissions from inland waters. Nature. 503(7476), 355-359

Sawakuchi, H.O., Neu, V., Ward, N.D., Barros, M.L.C., Valerio, A.M., Gagne-Maynard, W., Cunha, A.C., Less, D.F., Diniz, J.E., Brito, D.C., Krusche, A.V., Richey, J.E. (2017) Carbon dioxide emissions along the lower Amazon River. Frontiers in Marine Science. 4 (76) doi: 10.3389/fmars.2017.00076

Ward, N.D., Bianchi, T.S., Medeiros, P.M., Seidel, M., Richey, J.E., Keil, R.G., Sawakuchi, H.O. (2017) Where carbon goes when water flows: Carbon cycling across the aquatic continuum. Frontiers in Marine Science. 4 (7) doi: 10.3389/fmars.2017.00007.

Response: the estimation of global carbon cycle and budgets is constantly improved and perfected. Here, we want to express the important of aquatic system in the global carbon cycle.

We know the important of inland water $CO_2$ budget, the meaningful paper (Raymond et al., 2013) also was cited in our MS. We lose sight of the latest progress in this aspect (Sawakuchi et al., 2017; Ward et al., 2017). We should add the recent syntheses of global inland water $CO_2$ budgets in the introduction. Thus, we revised this part according to the reviewer's comment and added these

valuable references in the corresponding context.

5) Line 50: See previous comment on Line 20. The factors controlling C/N in terrestrial environments and inland waters are grossly oversimplified. C/N in inland waters is not only a result of OM origin. Molecules are selectively leached from soils during mobilization into streams (or even the flow paths that come before this such as throughfall, stemflow, etc). Molecules are also selectively degraded and sorbed/desorbed during transport, influencing composition. The review paper mentioned above is a good place to start for honing the conceptualization and discussion of inland waters.

Response: the influencing factors to the variation of POC/PON are very complex.

**Essential difference:** difference accumulation rate of C and N for different plant.

The organic nitrogen presents in protein and nucleic acid of plant preferentially. Thus the organic nitrogen content in higher plants is lesser than it in lower plant (such as algae). Because of that the lignin and cellulose, which includes low organic nitrogen content, are the main component in the higher plants (Giresse, 1994). Recent study indicate that C/N ratios of higher plants can research to 30, even more than 30 (Müller, 1999). However, C/N ratios of lower plants only research to 10, commonly smaller than 10 (Tyson,1995; Kendall et al., 2001).

**Environmental condition:** microorganism degradation, photodegradation

The difference of mineralization rate between OC and ON also will change the ratio of carbon and nitrogen (POC/PON). The loss rates of OC and ON varied with the temperature, composition of organic matter, dynamic characteristics of water (Stief 2007; Gälman et al., 2008; Gudasz et al., 2010; Sobek et al., 2014; Cardoso et al., 2014).

We should discuss the influencing factors to the variation of POC/PON, at least should add many references for each impact factors, although they are work as supporting role in MS. We revised corresponding context in the paper.

6) Line 80: After reviewing the list of data used, it is not surprising to see the lack of inland water discussion. There is one river dataset listed as far as I can tellâ˘A˘Tthe Ipswich and Parker rivers, 2 fairly small temperate rivers. The other inland water datasets are from 7 lakes. While the ocean dataset seems to be decently large, the attempt at a "global synthesis" of inland waters made here is non-existent. Where is the Amazon River, which makes up 20% of the freshwater flow to the ocean? How about the Congo River, the Ganges-Brahmaputra River, the Changjiang River, and all of the world's large rivers? Not to mention streams from different settings. I would recommend reading the following review from the 1980's that did a more comprehensive job than done here:

Meybeck, M. (1982). Carbon, nitrogen, and phosphorus transport by world rivers. Am. J. Sci. 282(4), 401-450

Response: in fact, our inland data includes 11875 couples of POC and PON and contains 253 lakes. The lake dataset listed in the supporting information includes many lake groups. The lake group contains many lakes, not just a single lake. We can open the online link if we are interested in it. In my mind, we did not collect all the inland water data for the analyses of variation of POC/PON and worked as contrast in this study. The analysis is validated if the representativeness of analyses dataset for the inland water can satisfy the variation range of 'global synthesis'. Of course, the representativeness will be increased with much more data was utilized,

and we should pay more attention to the collection of inland water data if we want to estimate the discharge flux to the ocean from rivers. We read the reference (Meybeck 1982, is a very good paper) recommended by expert (reviewer 2), the POC/PON in Prof. Meybeck study (range from 6.9 to 13) was added. This valuable reference also was added in the context to discuss the variation of

7) For this present study to be meaningful, the authors need to include the majority of robust datasets currently available in the literature. It appears the authors only used data that could be readily downloaded from websites, rather than making a true effort to mine the literature. They have ignored the entire body of inland water literature.

Response: We can't need to assemble all the inland water data for POC, PON and POC/PON, of course, more data is necessary for more representative analyses. We use the representative inland water data includes more than 11875 couples of POC and PON in 253 lakes and 15 rivers. The different types of eutrophic, turbid small, large, and shallow lakes are included in the dataset. Thus, we think that the dataset used in this study could represent the variation of POC, PON and POC/PON in inland water, and could help us to further reveal the relationship between POC and PON and deviations in POC/PON in ocean.

**Anonymous Referee #2**

1) This paper expanded the global marine dataset on POC and PON, including extending the range northward a few degrees of latitude, and produced many new insights or conclusions compared to previously published studies. It's also good to see freshwater data included, and got some evidence of variability in different lake data. Such as, the finding of high C:N at high northern latitudes (ms. Fig. 2) is as far as I know novel and more or less inverts the temperature-based conclusions of Martiny et al. (who showed C:N increasing with temperature). The ms. figure 7C is quite different from what Martiny et al. (2013) showed in their figure 4. These new insights or conclusions compared to previously published studies suggest that there still are some critical things we need to know to deepen our understanding of global patterns in linkages of C and N. The authors have performed a great service in assembling these data and this is important to extend current knowledge to a wider range of geography. This paper should be published and I offer the following specific comments or suggestions on ways to improve the manuscript.

Response: Yes, as expert (reviewer 1) said, there is much knowledge of variation in POC and PON hasn't been sufficiently studied. Our study also suggest that there still are some critical things we need to know and more research on the stoichiometry is needed although our study shows some new findings.

2) Title – What is meant by "variation pattern?" Suggest a more descriptive title would be something like "Global patterns in particulate and dissolved organic carbon and nitrogen in the global ocean and inland waters."

Response: We considered the title using ' Global patterns '. However, the inland dataset missed many data of POC and PON in inland waters, which also mentioned by expert (reviewer 2), although our data set can satisfy the variation range of "global synthesis" in inland water.

3) Figures – All of them are too small, which made it really hard to see what was going on with the data. Suggest converting each on to landscape orientation and then filling the entire page with it, or submitted each figure respectively.

Response: We will submit each figure respectively when we submit the revised edition.

4) The Abstract is adequate. but the means of (12.2±7.5) should be noted, mean value ± error or standard deviation?

Response: It means mean value ± standard deviation, we revised it in the MS.

5) The Introduction is okay but not very inspiring.

Response: we revised the introduction and added some information according to the expert's (reviewer 2) suggestion.

6) I believe the analysis of distance to land (Fig. 4) is by far the most extensive one yet. The detailed analysis method should be introduced in section 2.2, although '3) Offshore distance ranges...........' was mentioned.

Response: We added the some detailed description of data process in the section of method.

Offshore distance ranges (5, 10, 15, 20, 25, 50, 75, 100, 125, 150, 200, 300, 500, 800 and 1100 km) were created via buffers establishment module in Arcgis 10 (Esri) (Following figure). The buffers overlap with continent was erased by terrestrial vector data. The samples located on different ranges of buffers (different distance from offshore) can show the variation of POC, PON and POC/PON from coastal to open sea.

[Figure]

[Figure]

The establishment of buffers for different distance from offshore was implemented by Arcgis 10 (Esri). The amplification of regional part in United States West Coast ( USWC ) can clearly show the distribution of sampling points in each buffer.

7) The analysis concerning soil carbon and nitrogen is novel. However, there was no mention in the Methods as to where these soil data come from or how they were matched to the marine data.

Response: We added the detailed description of data process in the section of method.

8) There are some really intriguing patterns here that depart from previous work and which are based on what I believe to be the most comprehensive dataset yet assembled on these parameters although some imperfections should be polished. This dataset has some interesting patterns that will help us move stoichiometry forward.

Response: We very appreciate the helpful suggestion and comments from expert (reviewer 1). We carefully revised the MS according to the expert's comments.

[revised manuscript text omitted]

---

## Referee Report (RR1)

Review of "Variation pattern of particulate organic carbon and nitrogen in oceans and inland waters".

In this manuscript, Huang et al. compile and describe an updated dataset of POC and PON measurements from the ocean and inland waterways. They describe spatial and biophysical relationships in an attempt to explain variations in the dataset. There is clearly a lot of information present in this dataset and the authors do an exhaustive comparison of the data in many ways. However, I think the authors do not convincingly support their conclusions and do not put their work in a context where the reader can assess its importance. For instance, many of the values or relationships they cite as significantly different have overlapping standard deviations, and many of the significant relationships they describe are accompanied by figures that do not clearly show the relationships they describe. The authors present many relationships without identifying their significance to our understanding of organic matter production and consumption. I think a manuscript that clearly identifies new and statistically significant relationships and their importance would be a useful contribution to the field.

Specific comments:
1. The major conclusions appear to be:
   a. The global average POC/PON from this study is higher than the Redfield ratio.
   b. The relationship between POC and PON in the northern hemisphere is different (higher regression slope) than in the southern hemisphere.

I do not find support in the text for the first conclusion. The POC/PON ratio you describe has an uncertainty that easily overlaps the Redfield ratio, not to mention published uncertainty in the Redfield ratio. It is not clear that your new value represents a new understanding or simply a different subset of data. Perhaps focusing your analysis on the one latitudinal region that appears significantly different than Redfield (80-90 deg N, Fig. 2), would be a worthwhile approach.

I believe your second point is derived from Figure 2, and at multiple points in the text you describe the latitudinal dependency of POC/PON as much higher in the northern hemisphere than in the southern hemisphere (lines 142-3). It is difficult to draw this conclusion from your figures. If you plotted POC/PON as a function of latitude (not forcing the separation through latitude bins) I do not think a relationship would emerge. Certainly there are only ~2-3 latitude ranges with POC/PON ratios that do not overlap the Redfield ratio at the 25th percentile.

2. Distance from shore:
       There does appear to be a difference between PON within 50km of shore in the northern hemisphere and further than 50km from shore. Is it possible to distinguish terrestrial inputs from coastal productivity? Perhaps you can expand on this finding.

Other comments:
Line 15: "some new points" – it is unclear what this means
Line 23 and elsewhere: "morphology"? Do you mean size? Or does the shape of the lake impact its organic matter?

Line 24: "significantly" – you use the word significantly throughout the text when presenting two values that do not appear to be statistically distinguishable. 6.89 ±2.38 to 7.59 ± 4.22 does not appear to be a significant change.

Line 35: Over what time frame and due to what influences do you expect these changes?

Line 61: This larger range for the Redfield ratio appears to contradict your abstract and conclusions.

Figure 1: Perhaps showing the sample density (using a heat map?) on the global map would make it easier to see where the majority of the samples come from.

Figure captions throughout: "…with whiskers covering most of the data" – I don't know what this means, but it does not appear that the whiskers cover the majority of the data range for most samples.

Figures throughout: Many of your figures (for example Figure 4) present data in bins vs. POC or PON or POC/PON. Without grid lines or some straight reference it is difficult to distinguish any relationship between the data and the y-axis values. These plots often give the impression that no relationship exists. You do not show your regression lines in the plots, only in your Supplemental tables. This makes it very hard for readers to follow your chain of argument and believe your results.

---

## Author Response (AR2)

Review of "Variation pattern of particulate organic carbon and nitrogen in oceans and inland waters". In this manuscript, Huang et al. compile and describe an updated dataset of POC and PON measurements from the ocean and inland waterways. They describe spatial and biophysical relationships in an attempt to explain variations in the dataset. There is clearly a lot of information present in this dataset and the authors do an exhaustive comparison of the data in many ways. However, I think the authors do not convincingly support their conclusions and do not put their work in a context where the reader can assess its importance. For instance, many of the values or relationships they cite as significantly different have overlapping standard deviations, and many of the significant relationships they describe are accompanied by figures that do not clearly show the relationships they describe. The authors present many relationships without identifying their significance to our understanding of organic matter production and consumption. I think a manuscript that clearly identifies new and statistically significant relationships and their importance would be a useful contribution to the field.

Response: We very appreciate the reviewer for his suggestive and valuable comments. Those comments are very helpful for revising and improving our paper. We have studied the comments carefully and addressed all of the comment. The main corrections have marked in the revised manuscript and the responses on a point-by-point basis are given below.

Specific comments:

1. The major conclusions appear to be:

a. The global average POC/PON from this study is higher than the Redfield ratio.

b. The relationship between POC and PON in the northern hemisphere is different (higher regression slope) than in the southern hemisphere.

I do not find support in the text for the first conclusion. The POC/PON ratio you describe has an uncertainty that easily overlaps the Redfield ratio, not to mention published uncertainty in the Redfield ratio. It is not clear that your new value represents a new understanding or simply a different subset of data. Perhaps focusing your analysis on the one latitudinal region that appears significantly different than Redfield (80-90 deg N, Fig. 2), would be a worthwhile approach.

Response: Redfield ratio is usually defined as 6.63 in previous studies. The first conclusion is ambiguous according to the reviewer's comment. Here, we just consider the mean value of POC/PON, but not fully take into account of uncertainty in the comparison. We rewrote these ambiguous contexts and compared them separately according to the reviewer's suggestion. The significant difference between Redfield ratio and in situ observation in 80-90 ° N (low temperature) was focused in the discussion section (3.4.2 Temperature).

I believe your second point is derived from Figure 2, and at multiple points in the text you describe the latitudinal dependency of POC/PON as much higher in the northern hemisphere than in the southern hemisphere (lines 142-3). It is difficult to draw this conclusion from your figures. If you plotted POC/PON as a function of latitude (not forcing the separation through latitude bins) I do not think a relationship would emerge. Certainly there are only ~2-3 latitude ranges with POC/PON ratios that do not overlap the Redfield ratio at the 25th percentile.

Response: The difference of variation pattern in POC, PON and POC/PON between northern and southern hemispheres is embodied in two aspects, one is the latitudinal dependency

another is the couple relationships of POC and PON. The latitudinal dependency of POC/PON is insignificant in the low latitude area and southern hemisphere, but relative remarkable between 20° N ~ 90° N. The linear regression slopes between POC and PON in the northern hemisphere (including intercept, 7.06 and excluding intercept, 7.00) are much higher than in the southern hemisphere (including intercept, 5.97; excluding intercept, 6.03). We rewrote the relative context in the manuscript.

2. Distance from shore:

There does appear to be a difference between PON within 50km of shore in the northern hemisphere and further than 50km from shore. Is it possible to distinguish terrestrial inputs from coastal productivity? Perhaps you can expand on this finding.

Response: 50 Km could be used to distinguish the impact of land from ocean. We enhance the finding of the variation in POC, PON and POC/PON along the distance from shore.

Other comments:

Line 15: "some new points" – it is unclear what this means

Response: we rewrote this sentence. 'some new points' was removed in the context.

Line 23 and elsewhere: "morphology"? Do you mean size? Or does the shape of the lake impact its organic matter?

Response: it means the shape of lake. It commonly defined as the ratio of lake area to depth.

Line 24: "significantly" – you use the word significantly throughout the text when presenting two values that do not appear to be statistically distinguishable. 6.89 ±2.38 to 7.59 ± 4.22 does not appear to be a significant change.

Response: we revised the presentation in the context.

Line 35: Over what time frame and due to what influences do you expect these changes?

Response: much more data and higher accuracy model were used in the recent studies. These changes mainly caused by the estimation method.

Line 61: This larger range for the Redfield ratio appears to contradict your abstract and conclusions.

Response: Redfield ratio is usually defined 6.63 in previous studies. 6.63-7.7 is the range of POC/PON in the study of Redfield. We rewrote it to avoid causing ambiguity.

Figure 1: Perhaps showing the sample density (using a heat map?) on the global map would make it easier to see where the majority of the samples come from.

Response: we revised the figure to the intensity plot according to the reviewer's suggestion.

Figure captions throughout: "…with whiskers covering most of the data" – I don't know what this means, but it does not appear that the whiskers cover the majority of the data range for most samples.

Response: we revised these figure captions, and remove these sentences with ambiguous and unmeaningful description. The box plot can present statistical information for most samples.

Figures throughout: Many of your figures (for example Figure 4) present data in bins vs. POC or PON or POC/PON. Without grid lines or some straight reference it is difficult to distinguish any relationship between the data and the y-axis values. These plots often give the impression that no relationship exists. You do not show your regression lines in the plots, only in your Supplemental tables. This makes it very hard for readers to follow your chain of argument and

believe your results.

Response: The box plot could not be processed to regression lines. We replot some figures and added the regression lines in the plot.

[revised manuscript text omitted]

---

## Author Response (AR3)

Dear Associate Editor

We are grateful to the anonymous reviewer for his useful comments and also appreciate Associate editor for his patient work during the revise process of our paper.

We carefully revised the paper according to the reviewer's comments. We also re-polished the language of this paper and hope it can get much more readability for the readers. All the revisions were marked in the revised edition. The clear edition was also attached in the submission files.

The introduction sets up the motivation for the study well; however, there are a couple of sections in the manuscript that need appropriate use of the English language. Line 23: The sentence would read better if "etc. factors" were deleted. The first paragraph (Lines 36-41) in the Introduction needs to be edited for a clear and concise message. Line 38: What is meant by "subject"? Is subtract a better word? I have listed a few other corrections below.

Response: the language has been edited by a native English speaker. Some minor mistakes were revised according to the reviewer's comments.

Line 90: Are all the standard methods the same? It would be helpful to state that although there might be variability in methods results are comparable.

Response: 'standard methods' means that 'the C and N were analyzed by a C/N elemental analyzer'. The instrument model of C/N elemental analyzer and detailed pretreatment of sample were not completely consistent with each other. We revised the 'standard methods' to 'similar methods' and avoid misleading the readers.

Is the linear function excluding the intercept necessary?

Response: do you mean the 'linear regressions in northern and southern hemisphere (7.06 and 5.97)' in line 163? These slopes of the linear regressions include the intercepts. The linear function excluding the intercept was used to compare the linear function with intercepts and power function.

Why do the authors think that terrestrial impacts are weak from all distances form shore? Quick dilution? (Line 205)

Response: The impact of land is very significant in the northern hemisphere within 50 km from shore, but not significant (weak) in the southern hemisphere. The POC and PON is much higher (21.90 ± 11.01 μm/L and 3.19 ± 1.46 μm/L) close to the land than that (11.65 ± 3.58 μm/L and 1.67 ± 0.44 μm/L) off the land in the northern hemisphere. The weak terrestrial impact in the southern hemisphere could be caused by low land area and coastline. Many sampling sites far from the shore in the southern hemisphere also could reduce the terrestrial impact during the analyses process.

The higher DOC concentration is not obvious by comparing the two figures. For a clearer comparison, please provide a supplemental table with POC and DOC averages for groups

Response: We added the ratio of DOC/POC for each water type (1.48 ± 0.75 for Coastal water, 7.57 ± 6.83 for river, 1.18 ± 0.77 for lakes and 4.02 ± 2.13 for ocean) in the context.

line

Other comments

Line 65: Delete first "and" and place comma after "Nitrogen."

Response: this sentence was revised in line 65 'Nitrogen, light limitation and phytoplankton can only explain approximately 20% of .......'

Line 66: Add "and after organic matter.

Response: We added the 'and' after 'organic matter ', 'the temperature, composition of organic matter and dynamic characteristics of water will ......' in line

Line 77-78: Delete "is." Also, write complement and perfection in past tense (complemented and perfected).

Response: this sentence was revised to 'the elemental stoichiometry research of C/N in inland waters still need to be complemented and perfected'

Line 108: Delete POC and PON and refer to them as samples.

Response: POC and PON were deleted and this sentence was revised to 'The data in each latitudinal range include all ranges of temperature, time and depth for samples.'

Line 120: Delete "and so on" and replace with "etc."

Response: ' and so on ' was replaced by ' etc.' and the sentence was revised to ' These observational data were processed as lake groups, such as Great Lakes Group, Lacustrine Central Group, Alaskan Lakes etc. (Table S7).' in line 119.

Line 232: Replace "was" with "were" Also is location a better word than position?

Response: This sentence was revised to ' The lakes investigated in this study were sorted by the latitude according to the geographical position.' in line 241

Line 233: Replace "that in" with "the"

Response: 'that in' was replaced by 'the' in line 242

Line 274: Replace "small" with "smaller"

Response: small was replaced by smaller, and this sentence was revised to 'However, the variation of POC/PON (variable coefficient, 0.47) in river waters is much smaller than POC ' in line 285

Line 283: Did authors mean to say "closer" instead of "closing"?

Response: 'closing' was replaced by 'closer' in line 294 in 'bg-2017-68-manuscript-version5'.

Line 399: Delete "the"

Response: it was deleted and the sentence was revised to ' The correlation between POC and DOC is weak for individual water types, but is strong for whole data of lake, ' in line 408

Line 400: There is not a Fig. 9C. It seems like you are referring to 10C.

Response: Figure 9C only appeared one time in line 317 'POC/PON = 11.938*(land/coastline)$^{-0.078}$ ($R^2$=0.41) (Figure 9C)' in 'bg-2017-68-manuscript-version5'. We can't find the 'Fig. 9C' in line 400.

Is any misunderstanding?

Table S2 –S6. Please include the three mathematical functions: linear (with and without intercepts) and power functions in the table captions.

Response: We added the mathematical functions in the table captions, 'Table S2 Relationship between PON and POC for each latitudinal range. Three mathematical functions $POC=A_0\times PON+C_0$, $POC=A_1\times PON$ and $POC=A_2\times PON^{B2}$ were used to fit the relationship between PON and POC. The parameters and determined coefficients of each function are listed in the table. The $R^2$ with * marked is the best regression function for the POC and PON.' in supporting information.

Figures 2 and 3. Include A, B, C in the captions.

Response: We added the specific position of charts A, B, C, D in the figures 2 and 3 (in figures 2 and 3 of the revision edition).

Figure 10 C: Regression line would be clearer if it was darker.

Response: the regression line was replaced by black line in figure 10 (in figure 10 of the revision edition).